# Role of Participation in Activities and Perceived Accessibility on Quality of Life among Nondisabled Older Adults and Those with Disabilities in Israel during COVID-19

**DOI:** 10.3390/ijerph19105878

**Published:** 2022-05-12

**Authors:** Orit Segev-Jacubovski, Ephraim Shapiro

**Affiliations:** 1Department of Occupational Therapy, Ariel University, Ariel 40700, Israel; 2Department of Health Systems Management, Ariel University, Ariel 40700, Israel; eas97@caa.columbia.edu

**Keywords:** perceived accessibility, environment, participation in activities, quality of life, older adults, COVID-19, Israel

## Abstract

During the COVID-19 pandemic, quality of life (QoL) was reduced among many groups, including Israeli older adults. This study investigated perceived QoL, perceived accessibility of the living environment, and participation in activities among nondisabled older adults and those with disabilities in the community. It also examined whether the perceived accessibility’s effect on QoL occurs directly and/or indirectly via mediators of participation in community activities. A voluntary and anonymous survey was administered from February to May 2021 to 495 participants aged 60 and older. Respondents completed three questionnaires: WHOQOL-BREF, Community Integration Questionnaire-Revised (CIQ-R), and Perceived Accessibility of Living Environment (PALE). The main finding was that participation in activities in the community had a direct positive impact on QoL. Perceived accessibility of the living environment also had indirect positive effects on QoL through participation in activities in the community, for those without disabilities but, interestingly, not for those with disabilities. Hierarchal linear regressions revealed that participation in activities explained 53.3% of the variance for both groups while perceived accessibility added 1.1% for the nondisabled. We conclude that accessibility of living environment is a good indicator of positively perceived QoL through participation in various activities in the community for nondisabled older adults. This may be especially important during a pandemic.

## 1. Introduction

The number and proportion of people aged 60 years and older in the world’s population has been increasing at an unprecedented rate, one that is expected to accelerate in coming decades [1]. As of 2021, 12% of the Israeli population was older than 65 years [2], which is lower than the equivalent percentage in other developed countries. For example, in Japan, Italy, and Germany, it is 24% on average, and in Europe, the overall average is 17% [3]. Despite a low proportion of adults over 65 years of age relative to other Western and high-income countries, Israel has been innovative and successful in terms of its policies towards older adults. These include developing various social programs and infrastructure, instituting a strong network of support for community centers and informal caregivers, and facilitating employment among older adults in order to promote a healthier more productive and more engaged older population [4]. Moreover, Israel was ranked number 19 on the Human Development Index, which reflects human well-being [5]. Approximately 98% of older adults in Israel live in the community, in an environment familiar to them [3], because it increases their independence, freedom of choice, and self-control in daily life [6]. One’s advancing years is a phase of life rich with personal development and satisfaction for some older adults, whereas for others, it is a negative stage of life. The determinants of a good quality of life as people age can vary from one person to the next [7]. However, there is a need to better understand these factors and what can be done to improve quality of life among older adults. This is especially true during the period since the COVID-19 outbreak and its concomitant restrictions, which has had an impact on both physical and social resources. Public health and social measures, including physical distancing, avoiding crowded settings, and mask wearing, were implemented across the globe to suppress COVID-19 transmission and reduce mortality and morbidity [8].

*Quality of life* (QoL) reflects a subjective evaluation which is embedded in a cultural, social, and environmental context in relation to people’s goals, expectations, standards and concerns. In addition, QoL is affected by disease and health interventions [9]. Enhanced QoL includes elements of involvement in enjoyable and constructive activity in later life [10]. Moreover, successful aging is defined as high levels of involvement in physical, psychological, and productive activities, and social functioning in old age without major diseases [11]. However, during the COVID-19 pandemic, significant reduction was found in QoL in many countries [12,13,14], including in Israel [15,16], both among older adults [17,18] and people with disabilities [14]. At the same time, keeping daily routines during lockdowns reduced levels of stress, depression, and loneliness and enhanced QoL during the pandemic [16]. Activities affected by movement restrictions such as spending time in nature, exercising, walking, and supportive interpersonal interactions have been associated with enhanced well-being [19]. Working has been found to be associated with higher QoL, and participation in daily life activities has been found to be associated with both QoL and psychological distress, and to mediate the effect of psychological distress on QoL [16]. Specifically, healthy lifestyle behaviors, such as physical activity, have been found to be associated with higher QoL during the outbreak of COVID-19 among older adults [20]. However, recent research has found that participation in activities decreased during COVID-19 [14,16,19,21]. Reduction in movement and activities as well as the reduction in social interaction have been found to be associated with sleep problems and psychological disorders (e.g., stress, anxiety, depression) [22].

Environment is another element found to influence QoL during COVID-19. For example, proximity to large parks and numerous local facilities as well as lower neighborhood density, living further from the city center, and living in a larger dwelling were associated with better health and well-being outcomes for all ages in one study [23]. With increasing age, many older adults lose some of their abilities, become less mobile, and need help or devices to overcome their limitations. One of the most sensitive areas of negative impact is on the size of one’s *social space*, represented by the rate of use of shopping facilities, amenities, and social resources outside the household [24]. *Perceived accessibility* can be defined as the person–environment fit or the level of compatibility between one’s personal abilities (physical abilities/disabilities) and needs and the environment, which results in psychological well-being and better physical activity according to the ecological theory of aging [25]. During COVID-19, perceived accessibility was found to enable participation in daily life activities because unnecessary traveling by public transportation was restricted to prevent the spread of the disease [26]. Another study found that perceived accessibility to daily necessities and social activities helped people, especially older adults, to restore their mental health status even after the lifting of travel restrictions. Older adults’ social activities—essential for maintaining good mental health—were highly dependent on physical accessibility [27].

To the best of our knowledge, no other studies have examined the direct and indirect effect on the QoL of *perceived accessibility of the living environment* and participation in activities in the community during COVID-19 in Israel and none have examined older adults with disabilities compared to a nondisabled group. Understanding the interplay between individual participation in activities and environmental factors that influence older adult QoL will inform strategies to target the needs and preferences of community dwelling older adults and thereby improve their QoL. Therefore, the objectives of this study were: 1. to investigate the differences and correlations in perceived QoL, perceived accessibility of the living environment, and participation in activities in the community among nondisabled older adults and those with a disability; 2. to examine whether the effect of perceived accessibility on QoL occurs directly and/or indirectly via mediators of participation in activities in the community.

## 2. Methods

### 2.1. Procedure

This research was a voluntary and anonymous cross-sectional survey. This survey was undertaken during COVID-19 in Israel from February to May of 2021, a period that included social distancing measures such as avoidance of gatherings, wearing facemasks in indoor environments, and the closing of senior clubs [28]. The survey was disseminated by the investigators and by six experienced research assistants from a diversity of ethnic and geographic sectors, through social platforms such as WhatsApp and Facebook groups by sending an online survey link, by telephone interviews, or by face-to-face interview in public places. At the end of May, when seniors’ clubs started to open, interviews were also conducted there. The aim of the study was explained, anonymity and confidentiality were assured, and it was stressed that participation was voluntary, with no consequences if they refused. The participants gave informed consent through an online form, or written consent on paper, or by phone consent which was recorded. The participants were asked to complete four self-reported questionnaires: the WHOQOL-BREF, the Community Integration Questionnaire-Revised (CIQ-R), and the Perceived Accessibility of Neighborhood Environment Questionnaire (PAVE), as well as a demographic questionnaire. The survey was professionally translated into Arabic.

### 2.2. Participants

A convenience sample of 495 participants responded to the survey. The inclusion criteria were individuals aged 60 and older, Hebrew, or Arabic speakers, and agreement to participate. For the statistical analysis, the sample was divided according to participants who self-reported that they had a functional disability and participants who did not suffer disability referred to as “nondisabled older adults”. The study underwent an institutional review and was approved by the ethics committee of Ariel University (AU-HEA-OSY-20201217).

### 2.3. Measurements

The WHOQOL-BREF (Field Trial Version) [9]; the generic WHOQOL-BREF contains 26 items divided into four domains: physical health (7 items), psychological health (6 items), social relationships (3 items), environmental health (8 items), and overall perception of their health (2 items). Each item from the WHOQOL-BREF is scored from 1 (very dissatisfied/very poor) to 5 (very satisfied/very good). Questions 3, 4, and 26 are reversed. The mean score of items within each domain is used to calculate the domain score. Mean scores are then multiplied by four and a total WHOQOL-BREF score is calculated by mean score of the four domains ranging from 0 to 100, where 100 is the highest and 0 is the lowest QoL. In the current study, we omitted item number 21 about sex life due to the conservatism of the older Israeli public, especially the religious and Arab sectors. In the current study, the QoL items showed high internal consistency (Cronbach’s alpha = 0.870 in Hebrew, and 0.843 in Arabic).

The Community Integration Questionnaire-Revised (CIQ-R) [29], is an 18-item self-reported, standardized questionnaire, designed to assess an individual’s degree of community integration, meaning participation in activities. The instrument is divided into four subscales that measure, respectively: (1) home integration: active participation of the person in the activities of the home; (2) social integration: participation in a variety of activities outside the home and interpersonal relations; (3) productivity: involvement in employment, education, and volunteer activities; (4) electronic social networking: participation in electronic social networking. Most CIQ-R items were scored from 0 to 2: a score of 2 indicated that the individual performs the activity alone, score 1—someone else is also doing or might be doing the activity as interpreted in Hebrew (i.e., me and someone else is carrying out the activity or with the help of someone else), score 0—someone else does it for them. The total score (0–35) was calculated as well as subtotal scores: home integration (0–12), social integration (0–10), productivity (0–7), and electronic social networking (0–6). A higher score indicates a higher level of community integration [29]. Cronbach’s alpha coefficient of ≥0.70 was seen for the home integration subscale only, while for the total and the three other subscales, Cronbach’s alpha coefficients have been found to be low to moderate in a healthy Italian population aged 18–64 years old [30]. The CIQ-R was translated into Hebrew with the permission of the authors for the current study. In the current study, the total CIQ-R items showed moderate–high internal consistency (Cronbach’s alpha = 0.731 in Hebrew, and 0.760 in Arabic). Cronbach’s alpha coefficient for the home integration subscale was high (Cronbach’s alpha = 0.788 in Hebrew, and 0.826 in Arabic), and moderate for social integration (0.533 in Hebrew, and 0.298 in Arabic), productivity (0.651 in Hebrew, and 0.745 in Arabic), and electronic social networking integration (0.752 in Hebrew, and 0.447 in Arabic).

Perceived accessibility of living environment and services (PALE): this self-reported questionnaire was developed by the authors for the current research, see Appendix A. The purpose of the PALE is to subjectively estimate the extent of accessibility of the individual environment. The questionnaire includes nine items referring to: elevator or stairs, access paths to a building or house, access paths near the house, whether it is a flat area or has inclines or declines, lighting, signage, variety of services, number of services within walking distance, and number of services within a short drive. Each item was scored 0 (inaccessible) or 1 (accessible), and the number of services within walking distance scored from 0 to 2. The total score ranged from 0 to 10, with a higher score indicating higher perceived accessibility, and therefore a higher-level person–environment fit. Content validity was established by sending the questionnaire to five occupational therapists who were experts with elderly rehabilitation and accessibility, and two architects who were experts with the elderly and accessibility. In the current study, the PALE items showed moderate internal consistency (Cronbach’s alpha = 0.494 in Hebrew, and 0.619 in Arabic).

A sociodemographic questionnaire was used to elicit participant background characteristics: gender, age, years of education, marital status, person(s) with whom the respondent lived, nation/ethnic sector, and religion. Respondents were also asked to self-identify in one of four religious categories: secular, traditional, religious and very religious/Haredi (also called ultra-Orthodox), commonly understood categories in Israel, each distinguished by variant sets of beliefs and practices, and which have been associated with different types of health and well-being [31].

### 2.4. Statistical Analyses

Descriptive, bivariate, and multivariant statistical analyses were performed using Statistical Packages for the Social Sciences (SPSS) version 21 for Windows (SPSS Inc., Chicago, IL, USA). Significance level was set at *p* < 0.05. Pearson correlation coefficients were used to test the correlations between measures, and linear regression assumptions were tested; Kolmogorov–Smirnov was not significant, meaning the dependent variable of QOL had normal distribution and, no multicollinearity was found between the explanatory variables. Two hierarchal linear regressions were conducted to explain QoL: one for participants who reported that they had a disability and the other for participants without disabilities, referred to as “nondisabled older adults”. The variables were entered into the regressions in the following order: at step 1: age and gender; at step 2: education, religion, and nation; at step 3: three subscales of community integration which were correlated with QoL. Perceived accessibility of living environment was not entered into the regression since it was not correlated with QoL. A mediation analysis [32] was then performed to assess indirect effects using the Sobel test. The mediation model was examined by testing the significance of the indirect effect of the independent variable (perceived accessibility) on the dependent variable (QoL) through participation in activities in the community.

## 3. Results

### 3.1. Descriptive Analysis of Variable Study

The study sample consisted of 495 participants with a mean age of 69.66 (SD 7.02), ranging from 60 to 100. The male to female ratio was 36.8%:62.4%. Table 1 presents demographic characteristics of participants by groups. Most participants answered by an online survey link, but 187 were interviewed by phones, face-to-face in public places, or in seniors’ clubs. In the sample, 407 responded in Hebrew, and 88 in Arabic. In addition, 102 participants reported that they had a disability, which was 20.6% of the sample with a mean age 71.8 (8.71), ranging from 60 to 97. The nondisabled older adult group included 393 participants, representing 79.4% of the sample and who were significantly younger than participants with disability, with a mean age of 69.1 (6.4), ranging from 60 to 100.

Significant differences were found in QoL, community integration and participation, and perceived accessibility of the neighborhood environment between participants with disabilities compared to nondisabled older adults. Participants with disabilities reported lower QoL, participation in activities in the community, and perceived accessibility of the neighborhood environment compared to nondisabled older adults, as presented in Table 2.

Among older adults with disabilities, significant moderate positive correlations were found between QoL and total participation in the community (r = 0.375, *p* < 0.001), social integration (r = 0.401, *p* < 0.001), and productivity (r = 0.348, *p* < 0.001), and a weak correlation was found for electronic social networking (r = 0.298, *p* < 0.001), but not with perceived accessibility. For nondisabled older adults, significant moderate positive correlations were found between QoL and total participation in the community (r = 0.356, *p* < 0.001) and electronic social networking (r = 0.377, *p* < 0.001), and weak correlations were found with social integration (r = 0.279, *p* < 0.001) and productivity (r = 0.264, *p* < 0.001). In addition, a significant weak positive correlation was found between QoL and perceived accessibility (r = 0.277, *p* < 0.001). The correlations are presented in Table 3.

### 3.2. Regression Analysis for Explaining QoL by Groups

Two hierarchal linear regressions are presented in Table 4. The first regression explained QoL for older adults with disabilities. The second step was significant, although only the effect of education was significant, accounting for 12.4% of the variance (R^2^ = 0.129, F(3, 93) = 3.901, *p* < 0.01). The third step was significant, but only the effect of social activities and productivity in the community was significant, accounting for 51.5% of the variance (R^2^ = 0.661, F(3, 90) = 22.388, *p* < 0.001), indicating that social activities in the community enhanced better QoL. The second regression explained QoL for nondisabled older adults and showed that age and gender accounted for 2.8% of the variance (R^2^ = 0.028, F(2, 384) = 6.551, *p* < 0.01), with younger age and being female increasing QoL at the first step. The second step was significant, accounting for 12.3% of the variance (R^2^ = 0.151, F(3, 381) = 14.678, *p* < 0.001). In addition, higher education and religiosity were significant, accounting for 12.9% of the variance (R^2^ = 0.129, F(3, 93) = 3.901, *p* < 0.01). At the third step, participation in the community, social integration, productivity, and electronic social networking activities were significant, with education and Jewish nationality accounting for 53.4% of the variance (R^2^ = 0.685, F(4, 377) = 94.480, *p* < 0.001). At the fourth step, perceived accessibility of the neighborhood was significant, adding 1.1% of the variance above and beyond participation in the community and personal factors (R^2^ = 0.696, F(1, 376) = 89.201, *p* < 0.001).

### 3.3. Mediation Analysis

The mediation model was examined for nondisabled older adults alone since correlation between QoL and perceived accessibility was not significant among older adults with disabilities. The mediation model was analyzed by testing the significance of the indirect effect of the independent variable (perceived accessibility) on the dependent variable (QoL) through participation in activities in the community as presented in Figure 1. The results for nondisabled older adults showed partial mediation of QoL by participation in activities in the community. The results revealed that higher perceived accessibility was associated with more participation in activities in the community (IV to mediator-path a) (b = 0.86, *p* < 0.000). Higher participation in activities in the community was positively associated with QoL (direct effect of mediator on DV-path b) (b = 0.50, *p* < 0.000). Higher perceived accessibility was associated with greater QoL (direct effect of IV to DV-path c’) (b = 1.50, *p* < 0.000). The indirect effect of perceived accessibility on QoL through participation in activities in the community was significant (path ab) (b = 1.06, *p* < 0.000). Differences between the two last b-values (0.43) were significant according to Sobel’s formula (z = 4.12, *p* < 0.000), although the mediation was partial.

## 4. Discussion

### 4.1. Qol, Participation in Activities in the Community, and Perceived Accessibility

QoL is an important concept for older adults, one influenced by a variety of factors [9]. The current research expands the understanding of factors that influence older adult QoL. In this study, the main finding was that participation in activities in the community had a direct impact on perceived QoL. In addition, perceived accessibility of the living environment had an indirect impact through participation in activities in the community and some direct impact on QoL among nondisabled older adults. However, while participation in activities, particularly social activities and productivity in the community, enhanced QoL for older adults with disabilities, perceived accessibility of the living environment did not, in contrast to the nondisabled group.

We hypothesize that the possibly surprising differences between the disabled and nondisabled segments of the sample regarding their environments may stem from differential impacts of the COVID-19 period. Due to prior experiences, those with disabilities may have already adjusted to limitations in their physical environment and/or developed a support system. Participants with functional disabilities likely enjoyed the support and alternatives to external adaptation for inaccessibility familiar to them before COVID-19, such as overcoming inaccessibility of the living environment by transportation directly from their home to activities in the community. Previous research found that community-dwelling older adults with disabilities achieved successful aging by using adaptation and coping strategies, such as accepting help, and therefore relied on others to align their perception of successful aging with their experiences [33]. Moreover, support that older adult participants received from family and friends was likely more important for enabling their participation than having accessible physical environments, is supported by prior literature [34]. However, for many of those in the nondisabled category, the limitations on use of the environment and activities during COVID-19, either because of regulations or fear of being infected, were something new, and they may not yet have adjusted to it. As such, there may a stronger connection between the environment and QoL for them during COVID-19.

### 4.2. Participation in Activities in Community and QOL

The findings from the current research support the claim that successful aging is a psychological adaptation process, which includes participation in activities in the community, encompassing social, physical, psychological, and productive dimensions [11,35]. In particular, the findings support the “activity theory of aging,” which suggests a positive relationship between social activity and life satisfaction in old age; the more one interacts with others or is exposed to the responses of others, the greater the opportunity for reaffirming specific role identities. Activities could be informal, formal, and solitary activities [36]. Similar to the current results, continued participation in formal activity, such as socially productive activities, improves quality of life in early old age [37]. Another explanation suggests that increases in physical activity are associated with increases in self-efficacy, which in turn, are associated with higher physical self-worth and fewer disability limitations leading to greater life satisfaction [38].

Consistent with the current results, previous literature has shown that people with disabilities have less diverse participation, stay home more, have fewer social relationships, and have less active recreation [39]. The participation needs of older adults with disabilities related to daily activities of personal care are generally met. However, social activities, interpersonal relationships involving leisure as well as other community life activities are not totally met for older adults with disabilities [40]. Studies in several countries have found that QoL and/or social activity to be greater among participants without functional disability than those with moderate to severe functional disability [41,42,43]. In addition, chronic disease self-management among older adults has been correlated with health status and directly related to quality of life [42]. Social interactions and a healthy lifestyle may prevent IADL disability due to cognitive decline [43], and social integration may prevent ADL limitations over time, particularly for the very elderly who have greater risks for functional decline [44]. In contrast, other research found only a weak relationship between QoL and social participation among older adults with physical disabilities [45].

### 4.3. Perceived Accessibility of Living Environment and Services and QOL

In the current research, perceived accessibility of the living environment had an indirect impact through participation in activities in the community and some direct impact on QoL among nondisabled older adults. Environments are highly influential on our behavior and our exposure to health risks (e.g., air pollution or violence), access to services (e.g., health and social care), and opportunities that aging brings [1]. Accessibility of the living environment, whether perceived or actual, and the ability to be active outside the home all contribute to the well-being and QoL of older adults [46,47]. The current results confirm previous studies. For example, it has been found that Europeans aged 65 or older typically report higher scores for perceived accessibility to services and sites associated with greater QoL [48].

In the current research, the impact of perceived accessibility of the living environment on QoL among nondisabled older adults was mediated through participation in activities in the community. This is consistent with previous studies showing higher level of participation in a variety of activities in an accessible environment. For example, within-neighborhood recreational walking was found to be positively related with proximity of recreational facilities, infrastructure for walking, indoor places for walking, and presence of bridge/overpasses connecting to services [47]. Moreover, environmental barriers in the neighborhood were found to cause people with disabilities who experienced food insecurity, and difficulties in accessing suitable food [49]. Older adults tend to leave their homes if they have a positive image of their environment, such as walkability. Higher levels of social participation have been observed for older adults who felt strongly that their neighborhood setting was appropriate for their lifestyle, with key resources perceived as accessible [50]. Interestingly, a direct negative effect between perceived accessibility and depressive symptoms has been found, one mediated by negative connections between satisfaction with social relationships and physical activity among a convenience sample of older adults in Israel [51].

Our results confirm the ecology theory of aging which claims that adaptation of a person to their environment and their alteration of the environment are a part of the process of human adaptation. High person–environment fit is important for good physical and mental health, well-being, and QoL among older adults [25]. The impact of personal factors has been well documented in previous research, although showing some inconsistencies [52,53]. In the current research, findings showed that younger age, female gender, higher education, religiosity, and Jewish nationality enhanced better QoL for nondisabled older adults. Only higher education enhanced better QoL for older adults with functional disability. Several issues limit the interpretation of the current findings. First, the study is limited by its cross-sectional design, which does not allow for prediction of a causal relationship between the variables. Second, all measures were obtained by participant self-report. Third, since we distributed the survey largely through social networks, which assumes that it was more accessible for specific segments of the healthy population, the sample may not be fully representative. However, despite the challenges of COVID-19, some participants in our study were recruited face-to-face as well as online, thus increasing the sample’s representativeness and generalizability to at least some extent. Additional strengths of this study are comprehensive questionnaires for each of the study’s outcome measures, some new or used in Hebrew for the first time, as well as a sample representing the diversity of the population in Israel in terms of religion and nationality during COVID-19, which to our knowledge, is unique. Further research might include longitudinal studies that track patterns of change in research variables across time, invaluable in identifying trajectories of QoL for older adults. An additional study may focus on the relationship of internet use with QoL for older adults and participation in community activities.

## 5. Conclusions

Enhancing older adult QoL is a challenging goal of our society. The significance of this study is by informing ecological strategies to modify participation in activities and the environment to specifically target the needs and preferences of community-dwelling older adults. The findings from the current study indicate that accessibility of the living environment seems to be a good indicator of positively perceived QoL, through participation in various activities in the community for nondisabled older adults. However, for older adults with disability, participation in productivity and social activities in the community seems to be a good indicator of positively perceived QoL. Therefore, we recommend that policy makers and professionals working with older adults should seek methods for enhancing both accessibility of the living environment and integrated participation in the community. Specifically, for older adults with disability, we recommend providing opportunities for social gatherings and activities in the community, encouraging engagement in activities tailored to their personal needs and abilities in order to improve their perceived QoL.

## Figures and Tables

**Figure 1 ijerph-19-05878-f001:**
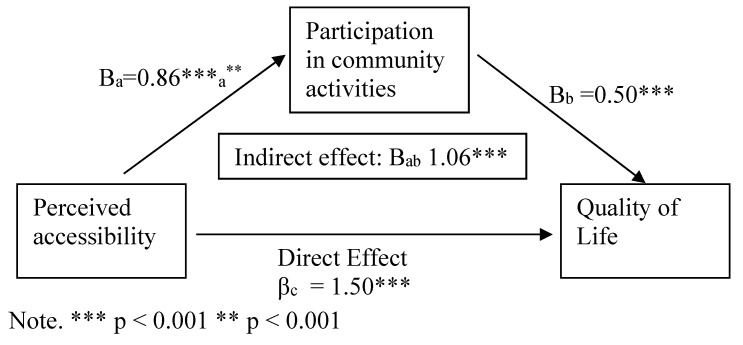
Regression analysis for mediation predicting quality of life. a. Independent Variable to Mediator; b. Direct effect of mediator on Dependent Variable; c. Direct effect of Independent Variable to Dependent Variable; ab. Indirect effect of Independent Variable on Dependent Variable through Mediator.

**Table 1 ijerph-19-05878-t001:** Participants’ demographic characteristics by groups (%).

	All *n* = 495	With Disability *n* = 102	Nondisabled Older Adults *n* = 393
Gender			
Female	62.4	63.7	62.1
Male	36.8	34.3	37.4
Marital status			
Married	76.2	63.7	79.4
Not married	23.8	36.3	20.6
Education			
Elementary school	21	43.6	15.2
High school and courses	23.1	27.7	21.9
Academic or certificate studies	55.9	28.7	63
Living with			
Alone	14.3	16.7	13.7
Intimate partner	75.7	58.8	58.8
Family member/s	7.7	15.7	5.6
Family member and formal caregiver	1.4	5.9	3
Formal caregiver	0.8	2.9	3
Nation			
Jewish	68.3	49	73.3
Arabic	31.7	51	29.5
Religion			
Secular	34.9	21.6	38.4
Traditional	26.5	30.4	25.4
Religious	32.3	42.2	29.8
Very religious/Haredi	6.3	5.9	6.4

**Table 2 ijerph-19-05878-t002:** Differences between older adults with functional disabilities and healthy older adults in research variables.

	All	With Disability	Nondisabled Older Adults	*p*
	*n* = 495 Mean, SD	*n* = 102 Mean, SD	*n* = 393 Mean, SD	
Total CIQ-R	18.22 (6.66)	13.26 (7.32)	19.50 (5.85)	0.000
Home integration	6.32 (3.04)	4.71 (3.47)	6.75 (2.77)	0.000
Social integration	5.89 (2.19)	4.62 (1.96)	6.23 (2.13)	0.000
Productivity	2.74 (1.97)	1.74 (1.92)	2.99 (1.90)	0.000
Electronic social networking	3.25 (2.12)	2.17 (2.10)	3.52 (2.03)	0.000
QoL Total	75.64 (10.89)	65.94 (9.77)	78.16 (9.70)	0.000
Physical health	72.93 (14.48)	57.36 (12.11)	76.97 (12.13)	0.000
Psychological	75.48 (11.54)	68.07 (12.20)	77.41 (10.55)	0.000
Social relationships	78.88 (14.54)	73.33 (0.79)	80.33 (13.89)	0.000
Environment	75.28 (13.26)	65.02 (12.15)	77.94 (12.20)	0.000
perceived Accessibility	7.09 (1.92)	6.01 (2.05)	7.37 (1.79)	0.000

**Table 3 ijerph-19-05878-t003:** Correlations between quality of life, accessibility, and participation in the community by groups.

QoL Total	Home Integration	Social Integration	Productivity	Electronic Social Networking	Total CIQ-R	Perceived Accessibility
Disability *n* = 102	0.190	0.401 **	0.348 **	0.298 **	0.375 **	0.104
Healthy *n* = 393	0.079	0.279 **	0.264 **	0.377 **	0.356 **	0.277 **

** *p* < 0.01.

**Table 4 ijerph-19-05878-t004:** Multiple hierarchical regressions for quality of life by groups.

	With Disability *n* = 99				Nondisabled Older Adults *n* = 387			
	B	SE	β	Adj. R^2^	B	SE	β	Adj. R^2^
Step 1				0.029				0.028 **
Age	−0.247	0.116	−0.215 *		−0.160	0.075	−0.108 *	
Gender	0.723	2.042	0.036		0.8732	0.988	0.146**	
Step 2				0.129 **				0.151 ***
Age	−0.151	0.114	−0.131		−0.104	0.072	−0.070	
Gender	0.522	1.996	0.026		1.028	0.965	0.052	
Education	4.394	1.221	0.376 **		4.131	0.707	0.323 ***	
Religious	1.213	1.076	0.108		0.916	0.473	0.093 *	
Nation	2.769	1.720	0.173		−1.466	0.860	−0.094	
Step 3				0.661 ***				0.685 ***
Age	−0.016	0.080	−0.014		−0.039	0.046	−0.026	
Gender	0.218	1.257	0.004		−0.566	0.657	−0.029	
Education	3.743	0.899	0.321 ***		2.025	0.518	0.158 ***	
Religious	0.545	0.677	0.049		0.075	0.295	0.008	
Nation	1.194	1.125	0.075		−1.684	0.530	−0.108 **	
Home integration					0.122	0.131	0.036	
Social integration	8.808	0.766	0.712 ***		10.064	0.426	0.721 ***	
Productivity	0.803	0.386	0.156 *		0.763	0.165	0.151 ***	
Electronic social networking	−0.402	0.409	−0.087		0.340	0.174	0.073 *	
								0.696 ***
					−0.061	0.046	−0.041	
					−0.539	0.646	0.027-	
					1.482	0.531	0.116 **	
					0.218	0.293	0.022	
					−1.482	0.525	−0.094 **	
					0.159	0.129	0.047	
					0.9989	0.419	0.716 ***	
					0.731	0.163	0.145 ***	
					0.333	0.171	0.071 *	
					0.619	0.169	0.117 ***	

*** *p* < 0.001, ** *p* < 0.01, * *p* < 0.05.

## Data Availability

The data presented in this study are available on request from the corresponding author. The data are not publicly available due to ethical restrictions.

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
