# Peer review of "Role of Participation in Activities and Perceived Accessibility on Quality of Life among Nondisabled Older Adults and Those with Disabilities in Israel during COVID-19"

_ijerph, 2022, doi:10.3390/ijerph19105878_

Round 1
Reviewer 1 Report
The authors investigated the correlations/associations of perceived QoL, accessibility of living environment, and participation in community among nondisabled and disabled Israeli older adults in pandemic period.
The study is comprehensive and relevant, though some corrections need to be made to improve its presentation.
Abstract
Lines 11-13 – There is repetition of words. Furthermore, the redaction of the objective is confusing. It must be same declared in the manuscript. Please, provide a better version.
Lines 21-24 (conclusion) – Here the authors must conclude in relation to its objective; what it is declared are some recommendations or may be a statement of some realities but is not sufficient as study’s conclusion.
Key-words - It is recommended not to use the same words of the title. Please try to select another one.
Introduction
- The authors need to follow the references format of the Journal, please correct lines 31, 34, 46.
- The paragraphs are very large (lines 44-63 and 64-89). It is necessary to restructure them.
- Why some parts are in bold? (Lines 38-43 and 87-89)
- The section “ 2. The present study”, must be removed.
- The lines 96-98 represent the purpose of the study I recommend moving it and placing it before the objective.
- Some ideas are repetitive in lines 83-98. I recommend the authors correct these sentences.
Methods
In subsection 3.3 there are many mistakes related to sizes and type of letter. Please review and correct in the entire manuscript (discussion also presented the same mistake).
Lines 149-151. It is confusing! Why are Italians declared here? It is confusing.
Lines 151-52: “The first author and Dr. Sonya Meyer translated the CIQ-R into Hebrew with the permission of the authors for the current study”. It is not necessary this detail, just inform the languages the questionnaires were applied.
As PALE questionnaire was developed by the authors it is recommend detailing the validation process; and of possible in supplementary material aggregated the questions.
3.4. Statistical Analyses. It is not clear if the variables are normal and if the lineal regression assumptions were tested.
It is recommended to detail a little more of the mediation analysis. For example, did the authors perform any test to assess the mediation analyses? Sobel test or other? Please mention these procedures here in the methods section and not just in results (lines 254-255).
Is it necessary a hierarchical analysis? It is not clear why. The disabled group present a small sample and these results must be interpreted carefully.
Results.
Tables 1 and 2: Please follow the same structure. The column “all” for example, in table 1 is at the end and in table 2 in the beginning.
Please change the 4.2 subsection title, it is not the best option. It says nothing.
Is it “really” necessary to divide this section? Please review all the subsection titles
Table 2. Please follow the same scale for the QoL dimensions in “all” column and in the groups (0-100?)
Lines 208- 215 the Table 3 information is repeated in text, please check if this tables is necessary. Also, provide information in relation the magnitude of the correlations (strong, weak…?).
Footnotes of the tables in general, please just let the signs or symbols that are declared in the table. It is not necessary to declare all the possibilities there.
Table 4 . Why is in footnote ªp= 0.088?
Discussion.
It is too big and repetitive. Please try to be more concise.
Do not let limitations in a special section. Is there any strength of this study?
Conclusion.
No observations.
Author Response
Thank you very much for the positive feedback and for taking the time to read the paper carefully. We appreciate all your comments which will help to greatly strengthen the paper.

Reviewer 2 Report
This is a well written and timely manuscript.
I only have minor comments:
Line 46, please fix "QoL effected" (e.g. "QoL is affected")
Line 59, please fix "and mediated between them" (not clear what this means)
Line 67-68, please fix "When age increasing" (e.g. "With increasing age")
Line 270, please fix "which found between" (e.g. remove "which found")
Author Response
Thank you very much for the positive feedback and for taking the time to read the paper carefully. We appreciate your comments which will help to improve the paper.

Reviewer 3 Report
- INTRODUCTION
Does the number of elderly people in Israel differ significantly from other countries?
How do the authors evaluate the policy towards the elderly in Israel compared to other countries?
During the course of the study, did the limitations of social contacts differ significantly from other countries at the same time?
Please indicate the place of Israel at Human Development Index or Social Progress Index and the relationship in this ranking with the quality of life.
Line 77 – no APA style: (Lawton & Nahemow, 1973).
- METHODS
How were people recruited for the research?
It can be assumed that people over 60 who use social media have a different quality of life than those who do not use such media. Because some people have been surveyed via the internet and some personally, these groups should be compared. In addition, in the limitations of research, it should be mentioned.
Please provide the reliability indicators for each language version of the questionnaire separately.
Table 4. - There should be a minimum of 10 people per factor in the regression analysis. There are only 99 people in the "With Disability" group and a very large number of factors thus the results are not entirely reliable.
- DISCUSSION
in the discussion Please include the results of the article replacement study
https://doi.org/10.3390/ijerph19074235
https://doi.org/10.3390/healthcare10040609
https://doi.org/10.3390/nu14061198
https://doi.org/10.3390/nu13124362
Author Response
Thank you very much for taking the time to read the paper carefully. We appreciate your comments which will help to greatly strengthen the paper.

Reviewer 4 Report
Please, see the attachment.

Author Response

(The authors gave the same response as above.)

Round 2
Reviewer 1 Report
The authors have improved consistently the manuscript. A hard job but you have done! I approve this final version.
In my opinion, it is not necessary to provide the PALE questions for this study, but if the authors intend that this survey be used in the future by other researchers who will evaluate perceived accessibility of living environment and services it is recommended to make it available in all the languages.
Author Response
Thank you very much. We provide the PALE questions in APPENDIX
Reviewer 4 Report
Authors did a great work and addressed the suggested revisions very well. The paper seems currently very clear, and with useful integrations and explanations. Some minor (really little) issues could however be solved, by my opinion, as follows.
- Table 1:
- following the explanation by authors, I would suggest to replace “Very religious” with “Haredi” also in Table 1.
- I see the indication that values are percentages (“%”) in the table, and not near the title. This aspect could be adjusted, e.g., as follows: “Table 1. Participants’ Demographic Characteristics by groups (%)”.
- Following IJERPH rules, I think that also info on Ethics Statements, Informed Consent Statement, and Data Availability Statement, could be added before the References in the manuscript, even though info on ethics and consent are already mentioned throughout the paper.
Finally, I would (generally) suggest authors to use track changes when they delete some parts, in order to make easier, for the reviewer, to check this. For the same reason, it could be better, by my opinion, to revise Tables in the manuscript, instead of providing new tables as supplementary material. However, this is only a formal aspect, that doesn’t impact the good work of the authors.
Author Response
Thank you very much. We appreciate your comments.
We wrote our answers in the attached file.
